The rule-based insensitivity effect: a systematic review

Kissi Ama ama.kissi@ugent.be
Harte Colin
Hughes Sean
De Houwer Jan
Crombez Geert
Department of Experimental-Clinical and Health Psychology, Ghent University , Ghent , Belgium
Boyes Mark
Electronic publication date: 2020 Jul 23
Publication date: 2020
Volume: 8
Electronic Location ID: e9496
Received 2020 Jan 27; Accepted 2020 Jun 17
Copyright: ©2020 Kissi et al.
Copyright year: 2020
Copyright holder: Kissi et al.
License: This is an open access article distributed under the terms of the Creative Commons Attribution License, which permits unrestricted use, distribution, reproduction and adaptation in any medium and for any purpose provided that it is properly attributed. For attribution, the original author(s), title, publication source (PeerJ) and either DOI or URL of the article must be cited.
License URL: https://creativecommons.org/licenses/by/4.0/

Keywords: Insensitivity effect, Rule-governed behavior, Rule, Instruction

Funding: Odysseus Group 1 grant awarded to Dermot Barnes-Holmes by the Flanders Science Foundation (FWO) Ghent University Grant BOF16/MET_V/002 awarded to Jan De Houwer This study was supported by Odysseus Group 1 grant awarded to Dermot Barnes-Holmes by the Flanders Science Foundation (FWO) and the Ghent University Grant BOF16/MET_V/002 awarded to Jan De Houwer. The funders had no role in study design, data collection and analysis, decision to publish, or preparation of the manuscript.

==============================
Background

Adherence to inaccurate rules has been viewed as a characteristic of human rule-following (i.e., the rule-based insensitivity effect; RBIE) and has been thought to be exacerbated in individuals suffering from clinical conditions. This review intended to systematically examine these claims in adult populations.

Methodology

We screened 1464 records which resulted in 21 studies that were deemed eligible for inclusion. Each of these studies was examined to determine: (1) if there is evidence for the RBIE in adults and (2) if this effect is larger in those suffering from psychological problems compared to their non-suffering counterparts. In addition, we investigated how (3) different operationalizations of the RBIE, and (4) the external validity and risks of bias of the experimental work investigating this effect, might influence the conclusions that can be drawn from the current systematic review.

Results

(1) Out of the 20 studies that were relevant for examining if evidence exists for the RBIE in adults, only 11 were eligible for vote counting. Results showed that after the contingency change, the rule groups were more inclined to demonstrate behavior that was reinforced before the change, compared to their non-instructed counterparts. Critically, however, none of these studies examined if their no-instructions group was an adequate comparison group. As a result, this made it difficult to determine whether the effects that were observed in the rule groups could be attributed to the rules or instructions that were manipulated in those experiments. (2) The single study that was relevant for examining if adults suffering from psychological problems demonstrated larger levels of the RBIE, compared to their non-clinical counterparts, was not eligible for vote counting. As a result, no conclusions could be drawn about the extent to which psychological problems moderated the RBIE in that study. (3) Similar procedures and tasks have been used to examine the RBIE, but their precise parameters differ across studies; and (4) most studies report insufficient information to evaluate all relevant aspects affecting their external validity and risks of bias.

Conclusions

Despite the widespread appeal that the RBIE has enjoyed, this systematic review indicates that, at present, only preliminary evidence exists for the idea that adults demonstrate the RBIE and no evidence is available to assume that psychological problems exacerbate the RBIE in adults.

The systematic review was registered in PROSPERO (CRD42018088210).

Introduction

Rules1 constitute a set of statements that can govern behavior in various domains such as personal, professional, social, and legal contexts. In most cases adherence to rules like “eat healthily if you want to live long,” “do not offend your boss,” “do not gossip about your friends,” and “do not drink and drive” is beneficial, in so far as doing so allows the individual to more readily obtain positive consequences (e.g., a long life, job certainty) or avoid negative ones (e.g., losing your friends, getting a fine). Yet despite the consequences of rule-following, rules can also continue to exert control over behavior even when they are no longer accurate. Within the behavioral-analytic literature, this pattern of behavior has been referred to as the “rule-based insensitivity effect” (RBIE) and has been defined as “an insensitivity of behavior to other contingencies2 due to rule-following” (see Kissi et al., 2018, p. 1).

To illustrate this effect more clearly, consider the following example. Imagine participants are asked to complete a learning task and are assigned to one of two groups: an instructions or no-instructions group. In both groups, they can initially earn points if they press the spacebar rapidly in the presence of a green square. Before starting the task, the instructions group is accurately informed about the contingencies operating in the task (i.e., that pressing the spacebar rapidly will cause them to earn more points). The no-instructions group, however, is not informed about these contingencies and thus has to figure out how to earn points via trial-and-error. About half way through the task, the task-contingencies are changed so that participants now have to press the spacebar slowly in order to earn points. Under such circumstances, it would be assumed that there is evidence for the RBIE if participants who were initially provided with accurate instructions, earned fewer points after the task-contingency change compared to those that did not receive such instructions (see Kissi et al., 2018 and Lefrancois, Chase & Joyce, 1988 for similar procedures).

Over the past decades, a number of studies have empirically examined the RBIE in the laboratory (e.g., Donadeli & Strapasson, 2015; Joyce & Chase, 1990; Miller et al., 2014; Ninness & Ninness, 1998). Elsewhere, applied researchers and clinical psychologists have appealed to this effect when attempting to understand and treat psychological suffering. For instance, it has been argued that the RBIE is at the core of various problems such as addiction, depression, and personality disorders (Baruch et al., 2007; Blackledge & Drake, 2013; Hayes & Gifford, 1997; McAuliffe, Hughes & Barnes-Holmes, 2014; Törneke, Luciano & Salas, 2008; Törneke, 2010). The idea here is that psychological problems are—amongst other things—the consequence of adherence to rules that reduce one’s ability to persist or adapt to what is required in a given situation (Blackledge & Drake, 2013).

Nevertheless, and despite the attention that rules and the RBIE have received, there is currently no systematic review available of the experimental work examining this effect. This is unfortunate, given that such a review is essential to draw general conclusions about the RBIE which can inform future research and clinical practice. Towards this end, we systematically reviewed the RBIE literature to examine if: (1) there is sufficient empirical support for this effect in adults, and (2) adults suffering from psychological problems display larger levels of this effect compared to those that do not suffer from these problems. We also investigated how (3) different operationalizations of the RBIE, and (4) the external validity and risks of bias of the experimental work investigating this effect, might influence the conclusions that can be drawn from the current systematic review.

Survey Methodology

Protocol and registration

The review protocol was designed in line with the PRISMA guidelines (Moher et al., 2009) and registered in PROSPERO (CRD42018088210).

Information sources and search strategy

To identify as many relevant records as possible, multiple electronic databases were searched (i.e., “Web of Science”, “PsychINFO”, “PsychArticles”, and “PubMed [Medline]”) using the search terms: “rule governed behavior”, “rule-governed behavior”, “rule governed behaviour”, “rule-governed behaviour”, “verbal regulation”, “instructional control”, “verbal rule”, “instructed behavior”, “instructed behaviour”, “instructed learning”, “instruction following”, “instruction-following”, “rule following”, and “rule-following.” These search terms were iteratively developed with experts on systematic reviews and rule-governed behavior, and were subsequently presented to other experts on systematic reviews and rule-governed behavior who were not associated with the project. All searches were conducted on 4∕10∕2017 by the first author (i.e., Ama Kissi) and yielded 1459 records. Five novel records were additionally retrieved by contacting experts in the field, which resulted in a final set of 1464 records that were assessed for eligibility.

Eligibility criteria

There were several general criteria that a record had to meet before being included in the current review: (1) it had to be a peer-reviewed journal article, (2) it had to be written in English, (3) it had to include a study that examined the RBIE by first asking participants to follow socially -or self-generated rules that initially corresponded with a set of contingencies but then became inaccurate after a contingency change, and (4) this study had to have an overall sample age of at least 18 years, (5) and at least 10 participants within each experimental group (see Van Ryckeghem et al., 2018 for similar eligibility criteria).

Furthermore, depending on the research objective under scrutiny, the individual studies reported in these records had to meet an additional number of criteria to be deemed eligible for inclusion. For instance, when addressing our first research question (“Is there evidence for the rule-based insensitivity effect in adults?”), we only included studies that did not focus upon individuals with clinical problems. That is, only studies which used convenience samples (e.g., students), samples taken from the general population, or those that were not diagnosed with clinical problems, or reported sub-clinical problems were included. Studies were deemed eligible for answering our second research question (“Do adults suffering from psychological problems display a larger RBIE compared to their non-clinical counterparts?”), if they used the following samples: individuals diagnosed with psychological problems (clinical group) or those who scored high on instruments measuring psychological problems but were not formally diagnosed with a clinical problem (sub-clinical group), and a comparison group consisting of individuals that did not suffer from the above problems or were recruited via convenience sampling.

Study selection process

Out of the 1,464 records that were assessed for eligibility, 1,446 were excluded because they were not published in English (n = 123), were not peer-reviewed journal articles (n = 207) (e.g., book chapters, dissertations, or conference papers) or dealt with a topic that did not meet our inclusion criteria (n = 1044). Three journal articles were, furthermore, omitted because they did not provide sufficient information to assess their eligibility. An additional 69 journal articles were excluded that were on the RBIE but were non-experimental (n = 6), relied on non-adult samples (n = 14), used samples with less than 10 participants per experimental condition (n = 41), or did not include a contingency change or manipulate accurate rules (n = 8). This resulted in a remaining total of 18 records consisting of 22 individual studies. One of these studies was subsequently omitted because it did not have at least 10 participants within each experimental group. As such, 21 studies were finally included in the systematic review. The eligibility of all studies were independently assessed by the first two reviewers (i.e., Ama Kissi and Colin Harte) initial agreement = 99% [kappa = .98], agreement after discussion = 100% [kappa = 1.00]). See Fig. 1 for the flow diagram of the study selection process.

Figure 1 Flow diagram of the study selection process.

Qualitative synthesis: coding procedure and items

Certain characteristics of each of the 21 studies were independently coded by the first two reviewers (i.e., Ama Kissi and Colin Harte) (initial inter-reviewer agreement = 96%, inter-reviewer agreement after discussion = 100%). These characteristics involved the source, study, task, and sample characteristics. The source characteristics entailed the year in which the first author published the study and the country where s/he worked in when the paper was published. The study characteristics referred to the type of task, experimental design, procedure, and analytic method that were used to examine the RBIE. Furthermore, the task characteristics entailed whether a study reported the exact instructions or rules that were used, how these instructions or rules were delivered (orally versus written) or generated (self [i.e., by the rule-follower]-versus socially [i.e., by another person than the rule-follower]), the reinforcement schedules that were used, the required behavioral responses, the type of consequential stimuli that were used, whether the contingency change was (un)signaled, whether a description was provided of who the experimenter was, and whether the experimenter was present. Finally, the sample characteristics that were evaluated were the size and mean age of the sample, the ratio of males:females, and whether the sample was selected (i.e., from either a healthy, clinical or sub-clinical population, or the general population) or non-selected (i.e., a convenience sample). These characteristics were evaluated for each experimental group.

Quantitative synthesis: vote counting

To synthesize the quantitative results of the included studies, we used the vote-counting method. This method was chosen because not all studies reported effect sizes or information that could be used to calculate such estimates. According to the Cochrane Collaboration guidelines for systematic reviews, the best way to use the vote-counting method is by assessing whether the results of the empirical studies fall into one of two categories: “positive” or “negative” effects (see Deeks, Higgins & Altman, 2008). Positive effects refer to results that are in favor of the predicted relationship between the independent and dependent variable(s), whereas negative effects refer to outcomes that are in the opposite direction of what is expected. We only judged (or voted) whether a study had positive or negative effects if it included a comparison group (i.e., a no-instructions group). That is, a group that received the same treatment as the rule groups but was not asked to follow the instructions or rules that these groups had to follow. We applied this restriction because we argued that such a comparison group is necessary if a study wishes to draw conclusions about the extent to which certain rules or instructions are responsible for the observed effects. In doing so, performances in the comparison group would serve as a baseline of how people behave in the absence of these types of rules or instructions. As such, if a study did not include such a comparison group, we argued that its effects were unclear (i.e., there was insufficient information to cast votes).

The outcome data that were preferably used to cast votes were measures of the central tendency (e.g., mean, mode, or median) of participants’ responses, during all blocks after the contingency change. If a study, however, did not report participants’ performances during all blocks following the contingency change, but only during a fraction of the trials after this change, we limited our analysis to that data. In the unfortunate event that no data was provided that could be used to draw conclusions about the central tendency of participants’ responding after the contingency change, we relied on the conclusions that the authors formulated themselves (Cerutti, 1991; Torgrud, Holborn & Zak, 2006) [Experiments 1 and 2]). Finally, in all of the above cases, if there were multiple contingency changes we only considered participants’ responding after the first change. This was, specifically, done to prevent carry-over effects from influencing the interpretation of the results.

All votes were independently cast by the first two reviewers (i.e., Ama Kissi and Colin Harte) in the following manner (inter-reviewer agreement = 100%, kappa = 1.00). For the first research question (“Is there evidence for the rule-based insensitivity effect in adults”), study results were considered positive if evidence was found for the RBIE. That is, if participants did not adapt to a novel task-contingency or rule (i.e., if their behavior was still in line with the self-generated or socially-provided rule that was in place before the contingency change). Furthermore, study results were considered negative if one of three conditions were met. First, if a task-contingency was changed and participants’ behavior was now always in line with this novel contingency. Second, if a self-generated or socially-provided rule was altered, and participants’ behavior was now always in accordance with this novel rule. Third, if both a task-contingency and rule was changed, and participants’ behavior was now always in line with this novel contingency and rule.

To cast votes for the second research question (“Do adults suffering from psychological problems display a larger RBIE compared to their non-clinical counterparts?”), we first assessed whether there was evidence supporting the RBIE. This was achieved in the same way as outlined above. If evidence for the effect was found, we subsequently examined if it was larger (in absolute terms) in the (sub-)clinical groups, compared to their non-clinical counterparts. If this was the case, then the study results would be categorized as positive. If these results were in the opposite direction, we would categorize them as negative.

Assessment of risks of bias

We, additionally, scrutinized the internal validity of the included studies. This examination involved assessing risks of bias using the Cochrane Collaboration tool for assessing risks of bias (Higgins & Altman, 2008) and the Office of Health Assessment and Translation (OHAT) Risk of Bias Rating Tool (NTP, 2015). Risks of bias can be defined as those aspects of a study design that can distort the conclusions that can be drawn from it. For the present review, we evaluated five potential risks of bias: selection, exclusion, performance, detection, and reporting bias. Note that these biases do not cover all risks of bias that are described in the Cochrane Collaboration and OHAT risks of bias tools. Indeed, given that these tools were not originally developed for assessing risks of bias in experimental-behavioral research, we selected and reformulated those risks of bias that we deemed relevant for evaluating such work.

For each of the studies, judgments of risks of bias (coded in terms of ‘high’, ‘low’, or ‘unclear’ risk of bias) were made in the following ways. To examine the possibility that there were systematic differences between the baseline characteristics of the groups that were compared (i.e., a selection bias), we examined: (1) the adequateness of a study’s sequence generation procedure, (2) whether the experimental group to which participants were allocated to was concealed, (3) participants’ past experiences with the experiment, and (4) the possibility that they were misclassified to experimental groups. Furthermore, to assess the likelihood of an exclusion bias (i.e., systematic differences in the exclusion of participants from a study) we evaluated the possibility that there were systematic differences between groups with regard to the amount, nature, and handling of missing outcome data. To determine the risk of a detection bias (i.e., systematic differences between groups in how outcomes are determined) we evaluated: (1) the validity and reliability of the outcome assessment methods, (2) the adequateness of the outcome assessments, (3) the adequateness of the methods that were used to determine sample sizes and (4) the adequateness of the methods used to analyze the results. Judgments concerning performance biases (i.e., systematic differences between groups in how they were treated or exposed to factors other than the manipulation of interest) were made by examining whether: (1) the experimental contexts were standardized, (2) participants were informed about the study objectives, and (3) researchers and/or participants were informed about the experimental group to which participants were allocated to. Finally, to assess the possibility of a reporting bias (i.e., systematic differences between reported and unreported findings) we assessed potential discrepancies between the outcomes that were specified prior to the study and those that were eventually reported.

Assessment of external validity

To determine the external validity of each of the included studies, we examined whether a study adequately described its eligibility criteria (in terms of age, sex, and diagnosis), the demographics of its sample, its study setting, its recruitment procedure, and the experimental manipulations that it used per experimental group.

Results

Summaries of included studies

For more information about the included studies, see Appendix S1 which contains summaries of all the included studies. These summaries are structured according to those studies that were deemed eligible to address the first (k = 20) and second research question (k = 1). There are two points worth noting about these summaries. First, they only include descriptions of those results that were relevant for the current research questions. As such, these summaries may contain less information than provided in the original study reports. Second, whenever it is mentioned that there is a difference between groups, this denotes an absolute and not a statistically significant difference.

Qualitative synthesis: source, study, task and sample characteristics

Source characteristics

The majority of the studies were written by a first author who did not work in the USA at the time of publication (i.e., Belgium [k = 3], Canada [k = 4], France [k = 2], Norway [k = 2], Switzerland [k = 1], USA [k = 9]) and most studies were published in the 2000s (k = 12).

Study characteristics

In the majority of the included studies, participants completed a conditional discrimination task (k = 14). In all of the studies, participants were allocated to one of the experimental groups, and conclusions about the RBIE were drawn by comparing the performances between these groups after a contingency change. Most of these studies examined the RBIE by examining how rules affected adaptation to changes (k = 11) or reversals (k = 6) in the non-instructed task-contingencies. See Table 1 for an overview of the study characteristics for each included study.

Task characteristics

In each of the 21 included studies, a description was provided of the precise instructions or rules that were used. Seventeen of these studies reported how they manipulated their rules or instructions. In 16 of these cases, this was via written text (five of these studies also provided additional oral rules or instructions). The majority of the studies used socially-generated rules (k = 19; five of these studies also used self-generated rules), intermittent reinforcement schedules (k = 15; two of these studies also combined such schedules with continuous reinforcement schedules) and tasks that required simple discrete responses (k = 14; in two of these studies discrete choice responses were also required). In 18 out of the 21 studies, points were used as consequential stimuli which were often exchangeable for a monetary reward (k = 10 out of 18). Of those studies that specified the contingency change (k = 9), seven of them stated that it was unannounced. Only one of the studies provided a description of the experimenter. Seven studies provided information about the presence of the experimenter. Of those studies, five stated that s/he was not present during the experiment. See Tables 2 and 3 for an overview of the task characteristics for each included study.

Table 1 Coded study characteristics.

	Type of task	Experimental design	Procedure	Analytic method	
Baruch et al. (2007)	Conditional discrimination task	Participants were allocated to one of the experimental groups	Non-instructed task contingencies reversal	Conclusions about RBIE are drawn by comparing performances between groups after a contingency change	
Cerutti (1991)	Conditional discrimination task	Participants were allocated to one of the experimental groups	Instructed task contingencies reversal	Conclusions about RBIE are drawn by comparing performances between groups after a contingency change	
Cerutti (1994)	Conditional discrimination task	Participants were allocated to one of the experimental groups	Instructed task contingencies reversal	Conclusions about RBIE are drawn by comparing performances between groups after a contingency change	
Dixon, Hayes & Aban (2000)	Gambling task	Participants were allocated to one of the experimental groups	Non-instructed task contingencies change	Conclusions about RBIE are drawn by comparing performances between groups after a contingency change	
Haas & Hayes (2006)	Conditional discrimination task	Participants were allocated to one of the experimental groups	Non-instructed task contingencies change	Conclusions about RBIE are drawn by comparing performances between groups after a contingency change	
Harte et al. (2017 –Experiment 1)	Conditional discrimination task	Participants were allocated to one of the experimental groups	Non-instructed task contingencies change	Conclusions about RBIE are drawn by comparing performances between groups after a contingency change	
Harte et al. (2017 –Experiment 2)	Conditional discrimination task	Participants were allocated to one of the experimental groups	Non-instructed task contingencies reversal	Conclusions about RBIE are drawn by comparing performances between groups after a contingency change	
Hayes et al. (1986)	Conditional discrimination task	Participants were allocated to one of the experimental groups	Non-instructed task contingencies reversal	Conclusions about RBIE are drawn by comparing performances between groups after a contingency change	
Kissi et al. (2018)	Conditional discrimination task	Participants were allocated to one of the experimental groups	Non-instructed task contingencies reversal	Conclusions about RBIE are drawn by comparing performances between groups after a contingency change	
Kudadjie-Gyamfi & Rachlin (2002)	Distributed choice paradigm where reinforcement could be increased if participants minimized the delay between a choice and its outcome	Participants were allocated to one of the experimental groups	Non-instructed task contingencies change	Conclusions about RBIE are drawn by comparing performances between groups after a contingency change	
Lefrancois, Chase & Joyce (1988)	Task in which reinforcement was dependent upon button presses	Participants were allocated to one of the experimental groups	Non-instructed task contingencies change	Conclusions about RBIE are drawn by comparing performances between groups after a contingency change	
Monestès, Greville & Hooper (2017)	Conditional discrimination task	Participants were allocated to one of the experimental groups	Non-instructed task contingencies reversal	Conclusions about RBIE are drawn by comparing performances between groups after a contingency change	
Monestès et al. (2014)	Conditional discrimination task	Participants were allocated to one of the experimental groups	Non-instructed task contingencies reversal	Conclusions about RBIE are drawn by comparing performances between groups after a contingency change	
Otto, Torgrud & Holborn (1999 –Experiment 1)	Conditional discrimination task	Participants were allocated to one of the experimental groups	Instructed task contingencies change	Conclusions about RBIE are drawn by comparing performances between groups after a contingency change	
Otto, Torgrud & Holborn (1999 –Experiment 2)	Conditional discrimination task	Participants were allocated to one of the experimental groups	Non-instructed task contingencies change	Conclusions about RBIE are drawn by comparing performances between groups after a contingency change	
Shimoff, Catania & Matthews (1981)	Conditional discrimination task	Participants were allocated to one of the experimental groups	Non-instructed task contingencies change	Conclusions about RBIE are drawn by comparing performances between groups after a contingency change	
Souza, Pontes & Abreu-Rodrigues (2012)	Task in which participants had to generate three-digit sequences that met a variability criterion in order to receive reinforcement	Participants were allocated to one of the experimental groups	Non-instructed task contingencies change	Conclusions about RBIE are drawn by comparing performances between groups after a contingency change	
Svartdal (1989)	Task in which participants had to count clicks and insert the number of clicks that they thought they heard in order to receive reinforcement	Participants were allocated to one of the experimental groups	Non-instructed task contingencies change	Conclusions about RBIE are drawn by comparing performances between groups after a contingency change	
Svartdal (1995 –Experiment 2)	Conditional discrimination task	Participants were allocated to one of the experimental groups	Non-instructed and instructed contingency change	Conclusions about RBIE are drawn by comparing performances between groups after a contingency change	
Torgrud, Holborn & Zak (2006 –Experiment 1)	Task in which reinforcement was dependent upon participants’ pattern of key presses	Participants were allocated to one of the experimental groups	Non-instructed task contingencies change	Conclusions about RBIE are drawn by comparing performances between groups after a contingency change	
Torgrud, Holborn & Zak (2006 –Experiment 2)	Task in which reinforcement was dependent upon participants’ pattern of key presses	Participants were allocated to one of the experimental groups	Non-instructed task contingencies change	Conclusions about RBIE are drawn by comparing performances between groups after a contingency change	

Table 2 Coded task characteristics.

	Report of exact rules/instructions used	Rule-delivery	Rule-generation	Reinforcement schedule(s)	Behavioral responses	
Baruch et al. (2007)	Yes	Written	Socially-generated	Continuous	Discrete choice responses	
Cerutti (1991)	Yes	Written	Self-generated	Intermittent	Discrete simple and discrete choice responses	
Cerutti (1994)	Yes	Written	Self-generated	Intermittent	Discrete simple and discrete choice responses	
Dixon, Hayes & Aban (2000)	Yes	Written	Socially-generated	Intermittent	Discrete simple responses	
Haas & Hayes (2006)	Yes	Written and orally	Socially –and self-generated	Continuous and intermittent	Discrete simple responses	
Harte et al. (2017 –Experiment 1)	Yes	Unclear	Socially –and self-generated	Continuous	Discrete choice responses	
Harte et al. (2017 –Experiment 2)	Yes	Unclear	Socially –and self-generated	Continuous	Discrete choice responses	
Hayes et al. (1986)	Yes	Written and orally	Socially-generated	Intermittent	Discrete simple responses	
Kissi et al. (2018)	Yes	Written	Socially-generated	Continuous	Discrete choice responses	
Kudadjie-Gyamfi & Rachlin (2002)	Yes	Written	Socially –and self-generated	Continuous and conditional	Discrete choice responses	
Lefrancois, Chase & Joyce (1988)	Yes	Written	Socially-generated	Intermittent	Discrete simple responses	
Monestès, Greville & Hooper (2017)	Yes	Written and orally	Socially-generated	Intermittent	Discrete choice responses	
Monestès et al. (2014)	Yes	Orally	Socially –and self-generated	Intermittent	Discrete simple responses	
Otto, Torgrud & Holborn (1999 –Experiment 1)	Yes	Written and orally	Socially-generated	Intermittent	Discrete simple responses	
Otto, Torgrud & Holborn (1999 –Experiment 2)	Yes	Written and orally	Socially-generated	Intermittent	Discrete simple responses	
Shimoff, Catania & Matthews (1981)	Yes	Written	Socially-generated	Intermittent	Discrete simple responses	
Souza, Pontes & Abreu-Rodrigues (2012)	Yes	Written	Socially-generated	Continuous	Complex response (i.e., three-digit combinations)	
Svartdal (1989)	Yes	Unclear	Socially-generated	Continuous and intermittent	Discrete simple responses	
Svartdal (1995 –Experiment 2)	Yes	Unclear	Socially-generated	Intermittent	Discrete simple responses	
Torgrud, Holborn & Zak (2006 –Experiment 1)	Yes	Both	Socially-generated	Intermittent	Discrete simple responses	
Torgrud, Holborn & Zak (2006 –Experiment 2)	Yes	Both	Socially-generated	Intermittent	Discrete simple responses	

Table 3 Coded task characteristics.

	Consequential stimuli	Announcement of contingency change(s)	Description of experimenter	Presence of experimenter	
Baruch et al. (2007)	Points that were exchangeable for a monetary reward	Unclear	Yes	No	
Cerutti (1991)	Points that were exchangeable for a monetary reward and a tone	Unclear	No	Yes	
Cerutti (1994)	Points	Unclear	No	Unclear	
Dixon, Hayes & Aban (2000)	Chips that were exchangeable for extra credit points	Unannounced	No	No	
Haas & Hayes (2006)	Points that were exchangeable for a monetary reward	Unannounced	No	Unclear	
Harte et al. (2017 - Experiment 1)	Points	Unannounced	No	Unclear	
Harte et al. (2017 - Experiment 2)	Points	Unannounced	No	Unclear	
Hayes et al. (1986)	Points that were exchangeable for a monetary reward	Unclear	No	No	
Kissi et al. (2018)	Points	Unannounced	No	No	
Kudadjie-Gyamfi & Rachlin (2002)	Points that were exchangeable for a monetary reward and time delays	Unclear	No	Unclear	
Lefrancois, Chase & Joyce (1988)	Points that were exchangeable for a monetary reward	Unclear	No	Unclear	
Monestès, Greville & Hooper (2017)	Points	Unclear	No	Unclear	
Monestès et al. (2014)	Points	Unannounced	No	Yes	
Otto, Torgrud & Holborn (1999 - Experiment 1)	Points	Unclear	No	Unclear	
Otto, Torgrud & Holborn (1999 - Experiment 2)	Points	Unclear	No	Unclear	
Shimoff, Catania & Matthews (1981)	Points that were exchangeable for a monetary reward	Unclear	No	Unclear	
Souza, Pontes & Abreu-Rodrigues (2012)	Points that were exchangeable for a monetary reward	Unannounced	No	Unclear	
Svartdal (1989)	Unclear	Announced	No	Unclear	
Svartdal (1995 - Experiment 2)	Sounds and lights	Announced	No	No	
Torgrud, Holborn & Zak (2006 - Experiment 1)	Points that were exchangeable for a monetary reward	Unclear	No	Unclear	
Torgrud, Holborn & Zak (2006 - Experiment 2)	Points that were exchangeable for a monetary reward	Unclear	No	Unclear	

Sample characteristics

On average, 58 participants were included in the analyses (SD = 33 and range: 21–150). The mean age of participants was 20 (SD = .16) and the average number of females was 34 (SD = 25). Note, however, that these values were based on the two and six studies that reported the mean age and gender proportions of the samples that were included for analyses, respectively. Twenty out of the 21 studies used convenience samples, whereas only one study used students that were selected based on the presence or absence of sub-depressive symptomatology (i.e., Baruch et al., 2007).3

Quantitative synthesis: vote counting

To address Research Question 1 (“Is there evidence for the rule-based insensitivity effect in adults?”) votes were only cast for the 11 out of the 20 studies that included a no-instructions group as a comparison group. These votes indicated that the results of each of these 11 studies were positive. No judgments could, however, be made for the one study that was relevant for addressing Research Question 2 (“Do adults suffering from psychological problems demonstrate larger levels of the RBIE compared to their non-clinical counterparts?”), because this study did not include a no-instructions group. For an overview of the vote-counting results for both research questions see Table 4.

Table 4 Overview of vote-counting results.

Type of change	Experiment	Evidence for the RBIE	
Studies used to answer Research Question 1 (“Is there evidence for the rule-based insensitivity effect in adults”)	
Task-contingencies			
	Dixon, Hayes & Aban (2000)	+	
	Haas & Hayes (2006)	+	
	Harte et al. (2017 - Experiment 1)	Unclear	
	Harte et al. (2017 - Experiment 2)	+	
	Hayes et al. (1986)	+	
	Kissi et al. (2018)	+	
	Kudadjie-Gyamfi & Rachlin (2002)	+	
	Lefrancois, Chase & Joyce (1988)	+	
	Monestès, Greville & Hooper (2017)	+	
	Monestès et al. (2014)	+	
	Otto, Torgrud & Holborn (1999 - Experiment 2)	Unclear	
	Shimoff, Catania & Matthews (1981 - Experiment 1)	+	
	Souza, Pontes & Abreu-Rodrigues (2012)	+	
	Svartdal (1989)	Unclear	
	Torgrud, Holborn & Zak (2006 –Experiment 1)	Unclear	
	Torgrud, Holborn & Zak (2006 –Experiment 2)	Unclear	
Instructions			
	Cerutti (1991)	Unclear	
	Cerutti (1994)	Unclear	
	Otto, Torgrud & Holborn (1999 - Experiment 1)	Unclear	
Task-contingencies and instructions			
	Svartdal(1995 - Experiment 2)	Unclear	
Studies used to answer Research Question 2 (“Do adults suffering from psychological problems display a larger RBIE compared to their non-clinical counterparts?”).	
Type of change	Experiment	Evidence for a larger RBIE in the clinical group	
Task-contingencies			
	Baruch et al. (2007)	Unclear	
Notes.

+ indicates that there was evidence for the RBIE. - indicates that participants in the rule-group(s) adapted to the change in the task-contingencies or instructions. ‘Unclear’ indicates that there was insufficient information to cast votes.

Assessments of risks of bias

Most of the included studies did not report the necessary information to assess all relevant domains of risks of selection, performance, exclusion, and detection bias. Nevertheless, the following can be said about those study aspects that we could draw conclusions about. Of the eleven out of the 21 studies that used a no-instructions group as a comparison group, none assessed the possibility that this group followed similar rules as the rule groups during the experiment. As a result, it could be that in these studies participants were misclassified to experimental groups. That is, there remains a possibility that participants were inaccurately thought to belong to a comparison group while in fact their behavior was actually governed by rules similar to those manipulated in the experimental groups. Furthermore, for the remaining domains, we argued that there were low risks of bias. Indeed, we argued that there was a low risk of reporting bias, seeing as there was a correspondence between the outcomes that were specified prior to the study and those that were actually reported. With respect to standardization of the experimental contexts, we argued that there was a low probability that the experimental groups were treated differently (performance bias). We also argued that there was a low probability that the methods that were used to assess the study outcomes were invalid or unreliable, and that the experimental groups differed with respect to how these outcomes were assessed (detection bias). See Appendix S2 and S3 for an overview of the judgments that were made for each aspect or domain of a study that could lead to a risk of bias.

Assessment of external validity

The majority of those included studies that were relevant for examining our first research question (“Is there evidence for the rule-based insensitivity effect in adults?”) (k = 20) did not report all relevant demographics (i.e., mean age, sex, and education level) of their samples (k = 13) nor their recruitment procedure (k = 13). Most of these studies (k = 16), however, explicitly described the setting in which the experiment took place, and all of them provided a detailed description of the experimental manipulations per group. The one study that was relevant for examining our second research question (“Do adults suffering from psychological problems display a larger RBIE compared to their non-clinical counterparts?”), selected participants based on the presence or absence of sub-clinical symptoms of depression, reported the eligibility criteria that they used, the demographics of their sample, and the experimental manipulations per group. Nevertheless, this study did not provide information about the experimental setting nor the procedure used to recruit participants.

Discussion

Rule-following is an essential human ability which can allow people to contact certain consequences more quickly and efficiently. Yet it has been argued that, under some conditions, this ability can also undermine people’s sensitivity to other environmental contingencies (i.e., RBIE) and can lead to a wide range of clinical problems. Despite the presumed importance of this effect for our understanding of human behavior in general and human suffering in particular, to date, no systematic review has been carried out of the experimental work that has examined these claims. To this end, the present study systematically reviewed the RBIE literature to determine: (1) if there is evidence for the RBIE in adults and (2) if this effect is larger in adults suffering from psychological problems compared to their non-suffering counterparts. In addition, we investigated how (3) different operationalizations of the RBIE, and (4) the external validity and risks of bias of the experimental work investigating this effect, might influence the conclusions that can be drawn from the current systematic review.

Our results can be summarized as follows: (1) there is preliminary evidence for the idea that adults demonstrate the RBIE; (2) at present, there is no evidence to support the claim that psychological problems moderate the RBIE in adults; (3) similar procedures and tasks have been used to examine the RBIE, however, their precise parameters differed across studies; and (4) most studies did not report sufficient information to evaluate all relevant aspects concerning their external validity and risks of bias. In the following sections, we will elaborate on each of the above-described points and their implications for our understanding of this effect.

Evidence for the RBIE

Remarkably, only 11 out of the 20 studies that were deemed relevant for addressing our first research question (“Is there evidence for the rule-based insensitivity effect in adults?”) were eligible for vote-counting, because they included a no-instructions group (as a comparison group). Of these studies, the results showed that after the contingency change, the rule groups were more inclined to demonstrate behavior that was reinforced before the change, compared to their non-instructed counterparts. At first glance, this seems to suggest that when adults are asked to follow initially accurate rules, they experience more difficulties adapting to changes in contingencies (compared to when they are not asked to follow such rules). Nevertheless, the risk of bias assessments showed that such a conclusion may be premature because none of the 11 included studies assessed whether their no-instructions groups functioned as adequate comparison groups. That is, none of these studies examined if, during the experiment, participants in their comparison group did not follow rules about the task-contingencies that were similar to those followed by the rule groups. As a result, this made it difficult to determine whether the effects that were observed in the rule groups could be attributed to the rules or instructions that were manipulated in those experiments.

Despite the fact that we found preliminary evidence for the RBIE in all 11 studies, it is important to acknowledge that there might be variables that increase or decrease the likelihood of observing this effect. For instance, according to past work, the RBIE might be less likely to occur if the experimenter is not physically present (e.g., Kroger-Costa & Abreu-Rodrigues, 2012), participants are provided with inaccurate as opposed to accurate instructions before a contingency change occurs (e.g., Hojo, 2002), and if the consequences for behaving in line with the actual task-contingencies outweigh those of following the rule (Donadeli & Strapasson, 2015). Unfortunately, a systematic examination of potential moderators of the RBIE (besides the moderating impact of the absence/presence of psychological problems) was beyond the scope of this systematic review. Nonetheless, we deem such an examination vital as it might further our understanding of the robustness of this effect. As such, we recommend that future work systematically examines those variables that might decrease or increase the RBIE.

Psychological problems and the RBIE

Despite the key role that the RBIE has been argued to play in psychological problems, only one of the included studies was deemed relevant for examining this idea. However, given that this study did not include a no-instructions group, no judgments could be made about the extent to which evidence was found for the RBIE and whether psychological problems moderated this effect. This suggests that there is currently no evidence available to draw firm conclusions about the relationship between psychological problems and the RBIE in adults. Furthermore, even if we evaluated the peer-reviewed journal articles (n = 69) which examined the RBIE but were omitted because they: (a) used samples smaller than 10, (b) samples from non-adult populations, (c) used non-experimental designs, and/or (d) did not manipulate rules or include a contingency change, we still failed to identify many relevant studies. Indeed, such a revised search only resulted in an additional four studies: two studies that investigated the impact of sub-clinical depressive symptoms in adolescents (McAuliffe, Hughes & Barnes-Holmes, 2014) [Experiments 1 and 2]), one study that examined that of ADHD in children (Kollins, Lane & Shapiro, 1997) and another study that examined that of schizophrenia in samples smaller than 10 (Monestès et al., 2014). We, therefore, strongly recommend that more work is conducted on the relationship between the RBIE and psychological problems to better inform clinical theory and treatment.

When carrying out such work, researchers should also explore certain variables that could moderate this effect in clinical groups. For instance, it might be that clinical groups (e.g., arachnophobics) are more insensitive to contingency changes if they follow pathology-relevant (e.g., “If you want to remain alive, always avoid places where there could be spiders”) but not pathology-irrelevant rules (e.g., “to gain points press the blue button”). Likewise, it is possible that different clinical groups (e.g., people suffering from psychosis vs. depression) demonstrate different levels of the RBIE because of differences in the origins (generated by imaginary agents vs. self-generated) of the rules they follow. Another possibility is that variations in the elements of the rules (i.e., the described stimuli [all spiders vs. tarantulas], responses [avoiding spiders vs. attacking them], and contexts [all spider habitats vs. the basement]), might contribute to differences in how people suffering from similar conditions (e.g., arachnophobia) adapt to contingency changes. We believe that such an endeavor would be useful because it could aid clinicians in developing more targeted treatments.

Operationalization of the RBIE

Our coding of task and study characteristics revealed that although most of the included studies used similar tasks and procedures, the precise parameters that were involved often differed. Specifically, many studies used conditional discrimination tasks during which participants could initially gain points if they followed the rules they received from the experimenter. In most of these studies, the task-contingencies were subsequently altered after a number of trials so that the previously effective rules were rendered ineffective. To illustrate, consider Kissi et al.’s (2018) Matching-To-Sample (MTS) task. This task consisted of two experimental phases. On every trial, participants were presented with four images. One image—called the ‘sample stimulus’—was presented at the top of the screen and always consisted of three identical symbols or letters (e.g., TTT). Three other images—called the comparison stimuli—were presented at the bottom of the screen. One of these images had two symbols or letters that were identical to the sample stimulus (e.g., TT%; most-like comparison stimulus), another had one symbol or letter identical to the sample stimulus (e.g., T%%; moderate-like comparison stimulus), while the third had no symbols or letters in common with the sample stimulus (e.g., %%%; least-like comparison stimulus). During the first phase of the experiment, participants could obtain points if they selected the comparison stimulus that was most-like the sample stimulus. However, during the second phase of the experiment, the task-contingencies were changed. Now, participants gained points whenever they selected the comparison stimulus that was least-like the sample stimulus. To examine the RBIE, some participants were given instructions telling them how to gain points in the task, whereas others had to learn about the task-contingencies via trial-and-error. This task is a conditional discrimination task because reinforcement for responses was conditional upon the characteristics of the sample stimulus.

Critically, despite the fact that most included studies used similar tasks, the precise stimuli (e.g., tones vs. images) that were used, the point in time in which the contingency change occurred (e.g., after two vs. three blocks), and the study outcomes (e.g., latencies vs. rate or accuracy of responding) often differed between studies. Generally speaking, if reliable evidence is found for a phenomenon, such variations are often viewed as a potential advantage because they enhance the generalizability of a study’s findings. Yet given that, in our opinion, it is unclear whether the RBIE was adequately assessed in any of the included studies in this review, we believe that this idea cannot be applied to our findings (see the previous sections “Evidence for the RBIE” and “Psychological Problems and the RBIE”).

External validity and risks of bias

The results revealed that many studies did not report all relevant demographics of their samples, how they were recruited, if the contingency changes were announced, and if the experimenter was present during the experiment. In addition, no study provided sufficient information to assess all domains of potential risks of bias. Taken together, this suggests that the reports of the included studies did not provide sufficient information to evaluate all coding items assessing their external and internal validity. The lack of such information is particularly problematic in the context of systematic reviews because it limits the conclusions that can be drawn from it. As such, we strongly recommend that, in future work, researchers report all information about their study that may enable readers to more readily draw conclusions about its external and internal validity (see Schulz, Altman & Moher, 2010 for guidelines).

Other considerations

In many of the studies, there was the implicit assumption that when people were asked to follow accurate rules, their behavior would be exclusively governed by those rules, and that if this was not the case, their actions would be exclusively guided by the task-contingencies. We would argue that such a reasoning might be problematic for two reasons (for similar arguments see Hayes et al., 1986). First, previous work suggests that when humans are not provided with rules they rarely demonstrate purely contingency-shaped behavior. Instead, they often generate and use their own rules about how they should behave in a particular context, based on their (trial-and-error) experiences in that context (Rosenfarb et al., 1992; Shimoff, Matthews & Catania, 1986). Second, such an interplay between environmental contingencies and rules may have also impacted the behavior of the rule groups that were described in the reviewed studies. Indeed, a closer look at the results of these studies showed that when behavior was considered rule-governed, it was rarely ever the case that participants consistently stuck to the rules they were told to follow. Rather, the results suggest that participants sometimes engaged with the task in ways that were not specified by these rules. There could be two possible explanations for this finding. A first possibility is that these deviations from the rules were unintentional and as such reflected erroneous responding. A second possibility is that instances in which participants discarded the rules that they were told to follow, actually constituted intentional attempts to explore instead of exploit the task-contingencies (Berger-Tal et al., 2014).

If the latter possibility is valid as well as the possibility that rules governed the behavior of the no-instructions groups, then this might suggest that comparisons between instructed and non-instructed groups might not inform us about the effects of rule-governed vs. contingency-shaped behavior per se. Indeed, such comparisons might then rather inform us about the relative degree to which socially-provided rules vs. environmental contingencies and self-generated rules vs. environmental contingencies influenced the behavior of the instructed and non-instructed groups, respectively. Yet given that we could not assess the plausibility of this assertion in the current study, this idea remains speculative. We, therefore, recommend that future work examines its validity so that we can gain a better understanding of how the RBIE should be conceptualized (e.g., as an insensitivity of behavior to other contingencies due to a stronger reliance on socially-generated rules than environmental contingencies).

Finally, to the best of our knowledge, there is currently no consensus about how contingency insensitive and sensitive behavior should be measured. Indeed, if anything, the implicit assumption is that behavior is contingency insensitive if it is not in line with a contingency, whereas it is contingency sensitive if it corresponds with a contingency. We believe that although such operational definitions can be useful in some respects, they lack the precision that is needed to measure these behaviors in a uniform and unambiguous manner. Indeed, given the broad and descriptive nature of these definitions, much variation can exist between studies in how they measure contingency sensitive and insensitive behavior. We believe that, although this is not an issue per se, it can become problematic when one wants to draw general conclusions across studies. We, therefore, recommend that future work offers more precise operational definitions of contingency sensitive and insensitive behavior.

Limitations

Several factors should be taken into account when interpreting our results. First, to determine whether or not behavior was in line with a previously effective rule and/or a novel contingency we used a liberal criterion. That is, we considered participants’ behavior to be in line: (1) with a previously effective rule if they demonstrated behavior that corresponded with this rule on at least a few trials, and (2) with a novel contingency and/or rule if they always behaved in line with this contingency and/or rule. As a consequence, it possible that if a different criterion were used, other findings would have emerged. Second, we opted for vote-counting for our quantitative research synthesis, which unlike the standard meta-analytic approach does not provide information about the magnitude of the observed effects (Koricheva & Gurevitch, 2013). Nevertheless, to gain some insight into these effects, we conducted a random effects model meta-analysis using those studies that reported sufficient statistical information. This analysis was based on six studies including a total of 377 participants (i.e., Haas & Hayes, 2006; Harte et al., 2017 [Experiment 2], Kissi et al., 2018; Kudadjie-Gyamfi & Rachlin, 2002; Monestès, Greville & Hooper, 2017; Monestès et al., 2014). It revealed a significant effect size of .76 (Cohen’s d for independent samples; 95% CI [.41–1.12]; p < .001) indicating that participants had far more difficulties adapting to a contingency change if, prior to the change, they received a rule as opposed to no rule. Third, across all studies that were deemed eligible for vote-counting, preliminary evidence was found for the RBIE. This was surprising, given that, in general, the likelihood of observing the same effect across all studies in a systematic review is rather low (Thornton & Lee, 2002). Usually, when such an overrepresentation of positive effects is observed, it is assumed that this might be due to publication bias, i.e., journals’ preference for publishing positive over negative findings (Joober et al., 2012; Thornton & Lee, 2002). Publication bias is particularly problematic in the context of systematic reviews, because it can lead to an overestimation of the existence of a particular effect. Therefore, we recommend the reader to take this bias into account when interpreting the findings of our systematic review. Finally, we adopted pre-defined inclusion and exclusion criteria which inevitably limited the scope of the review and as such the potential conclusions that can be drawn from it. For instance, we only considered peer-reviewed journal articles that examined one instance of the RBIE and one potential moderator of this effect in adult populations. Similarly, we only included experiments with groups that contained at least 10 participants, which led us to discard naturalistic studies and studies that adopted a single-subject methodology.

Conclusions

For several decades now, the RBIE has been argued to play an important role in human behavior in general and psychological suffering in particular. Yet despite its widespread appeal, the results of this systematic review suggest that strong claims about its existence and role in psychological suffering are currently unsupported and thus far unwarranted. Indeed, at present, only preliminary evidence exists concerning the RBIE in adults and no strong evidence is available to draw conclusions about its role in the development and maintaince of psychological suffering in adults. We, therefore, recommend that more systematic research is conducted on the RBIE so that future work can better evaluate the relevance of this effect for our understanding of human behavior and psychological suffering.

Supplemental Information

Supplemental Information 1 Overview items PRISMA checklist

Click here for additional data file.

Supplemental Information 2 Summary of all the included studies

Click here for additional data file.

Supplemental Information 3 Judgement of the relevant domains of risks of bias for studies used to answer Research Question 1

Click here for additional data file.

Supplemental Information 4 Judgement of the relevant domains of risks of bias for studies used to answer Research Question 2

Click here for additional data file.

Supplemental Information 5 Systematic review contribution and rationale

Click here for additional data file.

Additional Information and Declarations

Competing Interests

Author Contributions

Data Availability

1 Within the behavioral-analytic literature terms such as instructions and rules are often used interchangeably. Yet it is important to note that they are descriptive and not functional-analytical terms, given that they did not emerge from inductive, functional-analytic research. As such, in the current manuscript we will use them interchangeably as a way to orient the reader toward a specific class of verbal stimuli.

2 These contingencies can refer to other contingencies in the environment as well as those specified by a rule.

3 Note that we did not use the schizophrenic patients group from the Monestès et al. (2014) study to address our second research question because it had fewer than ten participants within each experimental group.

The authors declare there are no competing interests.

Ama Kissi conceived and designed the systematic review performed the systematic review and meta-analyses, analyzed the data, prepared figures and/or tables, authored the paper, and approved the final draft.

Colin Harte performed the systematic review, analyzed the data, authored and reviewed drafts of the paper, and approved the final draft.

Sean Hughes, Jan De Houwer and Geert Crombez conceived and designed the systematic review, reviewed drafts of the paper, and approved the final draft.

The following information was supplied regarding data availability:

This article contains a systematic review; there is no raw data.

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
