# Peer review of "The rule-based insensitivity effect: a systematic review"

_PeerJ, doi:10.7717/peerj.9496_

## Round 0.1 · original submission · Minor Revisions

I have now received three reviews of your manuscript, and I would like to thank all reviewers for their valuable comments.

All reviewers have provided comprehensive feedback on the manuscript, and their reviews are appended below. I will not reiterate the comments here, other than to note that in general the required revisions are relatively minor.

Please ensure you address all reviewer comments; however, I believe that the following issues warrant particular attention when revising your manuscript:

1) Reviewer 3 raises an excellent point about the search terms. Please clarify the search terms and Boolean operators used to conduct the systematic searches.

2) Please clarify the vote counting procedure in the text as well as diagram.

3) I agree with Reviewer 3 that, if possible, meta-analysing the studies for which appropriate information is available would strengthen the manuscript. This would also help in addressing Reviewer 1's final comment regarding effect size.

·

Basic reporting

no comment

Experimental design

no comment

Validity of the findings

no comment

Additional comments

Dear authors,

Overall the research appears to be conducted in a systematic fashion and communicated in a similar fashion, which is good. I can see the value of this work for people interested in investigating the RBIE.

The feedback below I personally deem 'minor points' for you to take or leave as you see fit.

In abstract it is stated: “(2) there is no evidence to assume that psychological problems exacerbate the RBIE in adults” It is also concluded that the overall evidence base is a bit shaky due to methodological issues. Perhaps a technically more correct way to frame this conclusion is then - “(2) At present there is a lack of evidence to assume that psychological problems exacerbate the RBIE in adults”. I feel like this softer framing of the conclusion is more allowing for the potential for that there could be an effect there if more research is conducted in a more systematic fashion.

Doing the above would seem more in line with conclusion on lines 371-373 “We, therefore, strongly recommend that more work is conducted on the relationship between the RBIE and psychological problems to better inform clinical theory and treatment.”

Line 37-38. Usually people provide a page number with the reference when making direct quotes.

Line 40-52. I really liked how you provided the fully fleshed out clear and simple example in this paragraph as the quote that finished off the prior paragraph is a little vague. Nice work! I have a very minor suggestion that you can take or leave as you see fit, to slightly modify final sentence to be: “Under such circumstances, further behaviour indicating the rule-based insensitivity effect, is that participants provided with accurate instructions would be more inclined to stick to these instructions after the task-contingency change compared to when they did not receive such instructions.” I was left wondering if your example was based off an actual study, or if it was simply made up for explanation purposes. I could see how utilizing a specific simple example from a published study would have been useful here, if such an example exists. Actually, now I think about it, personally what I would have done is provided 1-3 fully fleshed out examples from published studies. This would have been in aid of 1. Helping explain what the RBIE is, 2. Helping the reader better understand methodologies used in this research space, and 3. Choosing some disparate examples to highlight how the RBIE has been applied to diverse rules and/or contexts.

Line 71. I assume “(3) how operationalizations of the RBIE” is instead meant to read something like “(3) how the RBIE has been operationalized in the literature”

Line 93. I liked that you had an exclusion criterion based on sample size. Even if the criteria was arguably fairly generous. Personally, I think 10 per group is too low for adequate confidence a study and I would have been more stringent, but that is just a personal opinion of mine and I am not suggesting any changes here.

Lines 148-151. I assume a result was also classified as ‘negative’ if there was no statistically significant relationship found/reported by the study? At present it is worded only as results that are in the opposite direction. Also, in Table 4 the evidence for the RBIE is listed as ‘positive’ or ‘unclear’. This does not seem consistent with the ‘positive’ versus ‘negative’ terms used in text and so I found it a bit confusing.

Lines 281-285. Statement: “These votes indicated that the results of each of these 11 studies were positive. Furthermore, the results of the binomial test revealed that the probability of observing such findings in a sample of 11 studies is significantly larger than what would be expected by chance (p < .001).” The binomial test seems a little redundant here if all 11 studies appraised were positive… of course it is going to be ‘significant’. Something more interesting is that there is not one study with negative results… I can’t help but wonder if there is some publication bias at play here? That is, you have got all 11 studies with positive results because of the tendency of journals to preference the publication of statistically significant results over non-significant ones. I am assuming the issue of publication bias will come up in the discussion. *Okay, doesn’t look like this was not talked about in discussion section despite seeming relevant.

Lines 285-289. The reader is left wondering if that single study with the clinical sample reported positive or negative results. It wouldn’t hurt to briefly mention that at this point in the text?

Line 310. Tables 5 and 6 are largely full of question marks. I am left wondering if they are absolutely necessary and perhaps it would be better to state what was found in text without the need for tables. It just seems like the tables don’t provide enough information to be justified over a brief in-text summary. Maybe I am wrong here, I am not 100% sure to be honest, I suppose you could argue that it is consistent with the other tables and keeps things more systematic. I just make this comment as something for the authors to consider. There are already a lot of big tables in the manuscript… so perhaps the article would read a bit ‘cleaner’ (so to speak) without tables 5 and 6 (some brief in-text summary would need to be added to account for their omission of course)?

Lines 390-402. When talking about ‘conditional discrimination tasks’ providing description of a specific example from one of the published studies would have been useful here, I think. This paragraph is very general/vague and so unnecessarily difficult to follow.

General comment: The reader was given the impression that many of the studies reviewed used quite basic lab tasks that might have questionable ‘external validity’. That is, in everyday life following rules exists within quite dynamic and changing environments and contexts… whereas it seems in the research on this topic the studies are so highly controlled that it remains questionable whether the findings in the lab generalize to more ‘messy’ real-life contexts? I’m not totally sure if this observation is a good one as per one of my above comments there was a lack of detailed description of any specific studies within this manuscript. Despite the general well-written and systematic nature of the way the article is written, I found this lack of providing extra detail on some of the studies reviewed a little strange, to be honest.

General comment: Something hinted at in the discussion is that the 'rule based insensitivity effect' seems such a broad phenomenon and it seems unlikely that it would exist in all contexts (despite an arguably high [i.e., publication bias?] reporting of the prevalence. Perhaps discussion around this could be expanded and made clearer. The use of some specific examples from published studies would probably be useful.

General comment: Reader was left wondering what the general 'effect size' associated with the RBIE is? Is this something that is associated with a large impact on behaviour? Or small? Or somewhere in-between? I note that it was mentioned in the paper that not all studies report effect sizes to the point it could not be systematically evaluated. However, a little bit of discussion on this in the introduction and/or discussion sections would be a welcome addition in my opinion to provide the reader a better sense of how potentially 'powerful' the RBIE supposedly is.

Warm regards,
Shane Rogers

Reviewer 2 ·

Basic reporting

This study was well reported. The language used was clear and unambiguous throughout. The Introduction was adequate in providing background/context. The literature was well referenced and relevant in that focused on rule-governed behaviour and the theoretical background to this. The structure / layout of the paper conformed to ‘PeerJ’ standards, and to the norms for Psychology. Figures/tables were relevant, good quality, well labelled and described.

Experimental design

This was a review paper which is within the scope of the journal. The research questions were well defined, relevant and meaningful and it was clearly stated how the review would fill what was identified as a knowledge gap. The review was rigorously conducted to a high technical standard. Methods were thorough and were clearly described with sufficient detail and information to replicate.

Validity of the findings

The impact of the findings was adequately discussed with respect to all outcomes. All relevant data/results were provided. Data seemed robust, based on statistically sound analyses. Conclusions were well stated, linked to the original research question and limited to supporting results.

Additional comments

A well conducted and reported piece of work.

Just a few minor details for correction:

Introduction
L49-52
‘Under such circumstances…instructions’
Why is ‘you’ (i.e., the reader) referenced here? Instead of talking about what ‘you’ might do, the authors should continue to refer only to the groups of pts in the example.
L59
Capital ‘H’ for ‘Holmes’
L99
‘That is, studies that used convenience samples…’
This is a sentence fragment. Amend.

·

Basic reporting

Clear and unambiguous professional English is used throughout. I highlight below a small number of typos/suggestions that might improve this:
line 31 healthy -> healthily
line 59 and 474 Barnes-holmes -> Barnes-Holmes
line 138 “the proportion of women it included” -> “the ratio of males:females”
line 382 tarantula’s -> tarantulas

All other aspects addressed.

Experimental design

Under Information Sources and Search Strategy:
I am assuming that alternative spellings of behavior (i.e. behaviour) were automatically searched by default? Although this may seem a trivial point to raise, I note that the authors explicitly state that, for example, “instruction following” and “instruction-following” were entered as distinct search terms; so if the search algorithms did not automatically search for the other, hyphenated variant when the first was entered, the same might also be true for alternate spellings of behavior.

Under Quantitative Synthesis: Vote Counting
This may just be due to my own failing, but I cannot work out how the vote counting method applied led to the inclusion of 11 studies in the quantitative synthesis as detailed in the Prisma diagram. Was it because the other 10 studies did not report effect sizes or similar outcome measures? So the studies that were deemed "Unclear" were not included? It would be useful to clarify this by putting a number to it in the main text as well as in the diagram.

Validity of the findings

For those studies where information such as "Presence of experimenter" was deemed to be unclear, might it be beneficial to contact the corresponding author of the studies, where possible, to obtain this information?

Additional comments

This is an interesting, timely and necessary paper. My one major suggestion for improvement before publication might be - the authors have identified as a limitation of the study that vote-counting does not provide information about the magnitude of the observed effects unlike the standard meta-analytic approach. The rationale for using this method is that not all studies provided information such as effect size. But would it not be worthwhile adopting the meta-analytic approach to obtain information about the magnitude of observed effects from those studies that did include this information?

---

## Round 0.2 · accepted · Accept

Thank you for addressing the comments of the reviewers. I am delighted to accept your paper for publication in PeerJ.

·

Basic reporting

Good, no further comment.

Experimental design

Good, no further comment.

Validity of the findings

Good, no further comment.

Additional comments

Dear authors,

In my initial review I qualified my suggestions with "The feedback below I personally deem 'minor points' for you to take or leave as you see fit.". Despite this I see you have engaged with the feedback and made a number of changes. I feel like the changes made have improved the manuscript, so kudos to you. Also kudos for being interested in making changes even when it is qualified that they are not perceived as a necessary 'requirement'. This suggests to me that you are passionate about your work and interested in presenting it in the best possible way that you can, it is a nice thing to see.

Congratulations on your paper, and all the best for the future,
Shane L. Rogers